# The distribution of technology induced job loss: Evidence from a population-wide study in Norway

**Bjørn-Atle Reme** [1,2,3]*, **Ole Røgeberg**[4], **Jonathan Wörn**[1], **Bernt Bratsberg**[4], **Vegard Fykse Skirbekk**[1,5]

**1** Center for Fertility and Health, Norwegian Institute of Public Health, Oslo, Norway, **2** Cluster for Health Services Research, Norwegian Institute of Public Health, Oslo, Norway, **3** Department of Health Management and Health Economics, University of Oslo, Oslo, Norway, **4** Frisch Centre, Oslo, Norway, **5** Department of Psychology, University of Oslo, Oslo, Norway

* bjorn-atle.reme@fhi.no

## Abstract

Globalization and automation are leading to skill-biased structural changes in the labor market, resulting in the polarization of employment opportunities. These shifts are raising concerns about growing earnings inequality and gender disparities, particularly in occupations characterized by routine cognitive and physical tasks. This study utilizes comprehensive individual-level data from Norway to analyze gender differences in the routine intensity of occupations. The findings reveal significant and growing gender disparities. These disparities are most pronounced among individuals with low socioeconomic status. The analysis further identifies increasing gender differences in educational attainment as the primary contributor to the growing gender differences. Our results highlight the role of educational inequality in driving labor market disparities, emphasizing the need for targeted policy interventions to address these gendered dynamics, particularly among lower socioeconomic groups.

## Introduction

In recent decades, skill-biased technological change has allowed firms to automate and offshore routine-intensive tasks. This process has resulted in the erosion of low-skilled and mid-level jobs, while simultaneously increasing returns to education [1–4]. These trends are expected to continue. According to a study from the Organisation for Economic Co-operation and Development (OECD) covering 32 member countries, 14% of jobs were classified as highly automatable [5]. The disappearance of routine-intensive occupations may not necessarily result in an increase in long-term unemployment, as it is also expected to generate new opportunities and product markets that could drive labor demand [6]. From a classical economics perspective, labor saving automation frees up labor for new tasks, though the reconfiguration of the economy may involve a slow and painful shift in the labor market structure [7,8]. In fact, a recent OECD policy-brief found that countries with higher investments in robotics experienced *higher* employment growth [9], but also heightened job instability among workers in roles requiring less formal education. This raises the question of whether a

**Data availability statement:** The data used in this study encompasses educational outcomes, income, employment records, health records and demographic information for entire cohorts of the Norwegian population. These data were made available on loan for research purposes. Other researchers may apply for access to the same sources - see "helsedata.no/en" and "https://www.ssb.no/en/data-til-forskning/utlan-av-data-til-forskere".

**Funding:** This work was financed by the Research Council of Norway through its Centres of Excellence funding scheme (project number 262700) and the project DIMJOB (project number 296297). The funders had no role in study design, data collection and analysis, decision to publish, or preparation of the manuscript.

**Competing interests:** We declare no competing interests.

continuation of these trends will serve to amplify or dampen pre-existing inequalities [10,11], such as differences between individuals with low and high education levels.

Several studies have identified men in occupations requiring less formal education as a group that has experienced stagnation or declines in both wage level and labor market participation [12,13]. This development has coincided with a decline in marriage rates, fertility, and health outcomes [13–15]. While the reasons behind these developments are largely unknown, it has stirred debates among policy makers and academics as to the role of technological development.

Norway is a small, open economy with a highly educated workforce, extensive social safety net, and a compressed wage structure with centralized wage bargaining and strong labor unions. These factors have been argued to promote the adoption of technological innovations by industry [16]. Over recent decades, Norway has experienced both an increase in employment within high-skill occupations and a rise in automation through the use of industrial robots, reflecting broader trends observed in other developed economies [17]. Recent evidence from Norway's manufacturing sector, which leverages import data on industrial robots, suggests that occupational groups particularly vulnerable to automation experience a decline in employment share and wages but also show an increase in unionization [18]. This suggests that labor unions may play a role in mitigating some of the negative impacts of automation. While previous studies primarily have focused on the negative impacts of automation on employment or inequality, less is known regarding the distribution of this risk across the population and to what extent we should expect it to dampen or amplify pre-existing socioeconomic differences. In particular, evidence on the development over time within subgroups and its association with other known risk factors of social marginalization, is scarce. We hypothesize that men with lower education, who are more likely to be in routine-based occupations requiring fewer social skills, are at an increasingly higher risk of automation. Such a trend could exacerbate existing social inequalities [19].

The study aimed to investigate how a widely used indicator of occupational automation risk is associated and aligned at the individual level with measures of socioeconomic status in a register based study of the full population of Norwegian 45-year-olds, and how this differs for men and women. Our specific objectives were to (i) examine and explain the development in gender-specific risk across birth cohorts, and (ii) to estimate the association with other risk factors of social marginalization, including educational attainment, income, family background and health measures.

## Methods

### Data and analytical sample

We used linked population-wide administrative register data, covering information from several national registries. The registers were linked by Statistics Norway and then deidentified before researchers were allowed access. The acquisition of the data was part of the data structure at the Center for Fertility at the Norwegian Institute of Public Health, where data sets from the different registers were linked and stored at a secure server hosted by the Services for sensitive data (TSD), University of Oslo.

In this study we used demographic information and family status (national population register), labor market history (employer and employee register), educational attainment (national educational database), and physician visits (Norway Control and Payment of Health Reimbursements database; KUHR). The data were accessed by the researchers on March 1st, 2020.

Our study period covers 2003–2018, as this was the period where we had population wide records of occupational codes. Given the considerable differences in early-career labor market trajectories and the timing of family establishment across education levels, we included individuals at age 45 in each year. This approach, choosing mid-career, minimizes the risk of misleading associations arising from such variations in timing, driven by education. Hence, our sample consisted of the full population of individuals born 1958–1973 who were employed in the year they turned 45 and living in Norway (N = 900,559).

To measure structural risk of automation at the occupation level, we use the routine task intensity (RTI) index suggested in Acemoglu and Autor (2011) [20]. This approach to measuring routine intensity has been applied extensively in the labor economics literature on the impacts of skill based technological change, starting with the seminal contribution of Autor et al. (2003) [21]. The measure is a theoretically informed indicator of structural economic risk, classifying occupations based on the tasks performed in it. In particular, the index captures the extent to which an occupation is characterized by routine tasks in the cognitive, manual or interpersonal domain. This reflects an economic theory of automation predicting that routine tasks are more easily automated, an assumption that has received extensive empirical support in work employing the RTI index [21]. Measurement of the content of job tasks were retrieved from the O*NET database. This database consists of a highly detailed description of tasks performed in an occupation. From the large set of measurements in O*NET, Acemoglu and Autor (2011) suggests a subset of items which describe these dimensions. To summarize these items to a one-dimensional RTI measure for each occupation, we used the method suggested by Lewandowski et al. (2017) [22]. Last, to ease interpretation, we standardized the RTI score distribution in the sample to be mean zero with a standard deviation of 1 (RTI z-score). For further details on the construction of the Routine Intensity Index (RTI), see Supporting Information ("**Constructing the routine intensity index (RTI)**" in S1 File).

The RTI is one of two commonly used measures used to assess the risk of automation, with the other being the Frey-Osborne index (FOI) [21,23]. Although these measures are strongly correlated (see S1 Table), we based our analysis on the RTI for the following reasons. First, the RTI is task-based – it assesses to what extent a job consists of tasks that could be automated – while the FOI is based on expert assessment of occupations. Hence, the RTI has a more robust theoretical foundation, as it allows for occupational scores to change with their task content [24]. Additionally, the RTI has done comparatively better in predictive tests [25], and has been used in an extensive literature on skill biased technological change [2,21,26–28] (see [29] and [30] for a description and discussion of the difference between these measures, and other alternative measures used in the literature).

The main analysis in this paper is based on the 2003 version of the RTI – the first year of our study period. However, in the Supplementary material we also present all main results using both the FOI and the RTI based on 2019 O*NET data to assess robustness of our conclusions against changes in task content over time. S1 Table shows that all these measures of structural change are highly correlated, and our main conclusions are therefore independent of the selected measure.

Other measures used as covariates for stratification in the analysis were the following:

**Education:** The highest level of completed education, available from the National Education Database. There are 9 levels within the ISCED classification, where level 9 is "unknown". In the analysis we either use 4 categories: primary (1–2), secondary (3–5), 1–4-year university education (6) and master and higher university education (7–8), or dichotomize by whether or not the individual had university level education or not, i.e., 1–5 vs 6–8, referred to as "low" and "high" education, respectively.

**Own income:** Own income was retrieved from the income tax register. We calculated birthyear- and gender-stratified income quintiles. In the main part of the analysis, we use a binary variable indicating whether the individual belonged to the lowest income quintile.

**Father income**: Father's income was retrieved from the income tax register. We calculated the father's income quintile at age 40, stratified on his birth cohort. In the main part of the analysis, we use a binary variable indicating whether the father belonged to the lowest income quintile.

**Marital status**: Marital status was retrieved from the official marriage register. This contains complete records of marriages in Norway since 1974. For the analysis we used a binary indicator for whether the individual was registered as married at age 45. Hence, separated couples, or couples that for other reasons did report to the register, were not captured by our measure.

**Childlessness**: Using the population register we created a binary indicator for whether the individual was childless at age 45.

**Musculoskeletal/psychological problems**: We used the register of primary care utilization (KUHR) to create a binary indicator for whether the individual had visited primary health care (general practitioner) for musculoskeletal (L-chapter symptoms or diagnoses) and/ or psychological (P-chapter symptoms or diagnoses) problems during the year when being 45-years old. These two chapters were chosen, as they are the most common reasons for both health problems and sick leave among 45-year-olds in Norway.

Table 1 provides summary statistics across covariates, excluding covariates based on creating groups based on the underlying distribution (quintiles for own income and parental income).

## Statistical methods

All analyses were conducted using R, version 4.1.2.

**Data curation and organization.** Our sample included all individuals born between 1958 and 1973 who were employed in the year they turned 45 and resided in Norway. The database contains all registered employer-employee relationships in Norway each year, hence covering all employed wage earners. Occupational RTI scores were assigned to individuals based on their main occupation in their employment records at age 45.

**Statistical models.** Our aim was to examine how the risk of automation at the occupational level was associated with gender and various measures of socioeconomic status. This was achieved by estimating RTI z-score in different bivariate and multivariate regression models, stratified by gender. Hence, with this approach we both estimate RTI z-scores, and assess to what extent RTI z-scores can be explained by our explanatory variables.

**Table 1. Summary statistic for covariates.**

| Covariate | Proportion |
|---|---|
| Female | 49% |
| Childless | 15% |
| Married | 57% |
| Primary education | 25% |
| Secondary education | 36% |
| Low university education (Bachelor's) | 29% |
| High university (Master's or or PhD) | 10% |
| Visit to doctor for psychological problem | 9% |
| Visit to doctor for musculoskeletal problem | 22% |
| Number of observations (N) | 900,559 |

First, to assess time trends, we estimated the year- and gender-specific average RTI z-scores using regression models. Year and gender were included as binary indicators, hence the approach is equivalent to the conditional means. Since we used binary indicators, the method is semi-parametric, not resting on strict assumptions related to functional form.

Second, we examined how the structural risk of automation (RTI z-score) was associated with our measures of socioeconomic status and health (see Data and analytical sample). These models also included birth cohort dummies to adjust for potential time trends.

Third, we performed an attribution analysis where we estimated how much of the increase in automation risk gender gap could be explained by the various covariates included in the regression models. This analysis was carried out in two steps. First, the gender-specific change in covariate levels over the period 2003–2018 was calculated. Then, to estimate the contribution, we multiplied these changes by the corresponding covariate regression coefficients from the multivariate model to estimate the relative impact of each risk factor to the evolving gender-specific risk of automation.

Finally, to provide a detailed description of the variation in occupational risk scores in the most current population of 45-year-olds in our data - birth cohort 1973 - we estimated the RTI z-score for each observed combination of covariate values, separately for each gender. The results were plotted, ordered from low to high, in a bubble scatter plot where the bubble size reflects the size of the covariate group. To examine how these patterns have evolved over time, we also reproduced the plot for the first birth cohort in our data (born in 1958).

## Ethical considerations

The ethical approval for this study was given by The Ethics Committee of South-East Norway (#2018/434). Based on Section 35 of the Health Research Act, participant consent was waived by the Committee. Data from the different registers were linked by certified researchers who only had access to encrypted personal ID-variable. Extensive measures were taken to maintain the security and confidential handling of the research data. Throughout the process of analyzing data and presenting findings, we are committed to preventing stigmatization and to upholding ethical principles of honesty and transparency in reporting our findings. To achieve this, we used neutral language and avoided portraying vulnerable groups in ways that might be perceived as burdensome. Additionally, the study does not report on group sizes that could potentially disclose information about individual statistical units, in accordance with the recommendations outlined in the Handbook on Statistical Disclosure Control [31].

## Results

### Gender specific time trends in routine intensity

Our data covers the 15-year period 2003–2018. We first examined the development in routine intensity across this period, separately for each gender (Fig 1). In the supplementary materials we also present the development in RTI z-score for different levels of each covariate used in the analysis (S1 Fig).

While there already was a substantial gender difference in standardized routine intensity in 2003 - the score was 0.13 [95% CI, 0.12–0.14] for men and -0.13 [95% CI, (-0.14)-(-0.12)] for women - the difference gradually increased up until 2018, to 0.18 [95% CI, 0.17–0.19] for men and -0.20 [95% CI, (-0.21)-(-0.19)] for women. In the Supplementary material we also present the average risk score using the Frey-Osbourne index (S2 Fig), RTI 2019 z-score (S3 Fig), and not z-standardized RTI scores (S4 Fig). It should be noted that when presenting time trends with the FOI or raw RTI-score, the time trend is downward sloping, indicating that the overall risk of automation is falling during the study period. However, the decrease is smaller for men, causing an increasing gender gap.

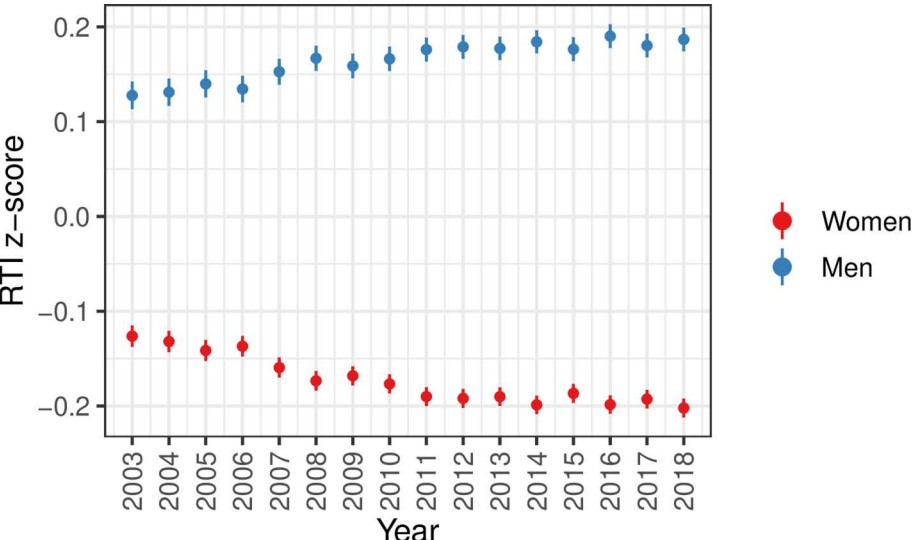

**Fig 1. The yearly average routine intensity index among 45-year-olds by gender.** The figure shows the yearly average RTI z-score, stratified by year, for 45-year-olds (birth cohorts 1958 to 1973) by gender. The 95% confidence intervals are indicated by the error bars. See S2 Table in the Supplementary Material for a corresponding table.

## The association between structural risk and socioeconomic status

In bivariate analyses, RTI z-score was most strongly associated with low education (education below university level) with coefficients of 1.11 (p < 0.001) for women and 0.82 for men (p < 0.001) (Fig 2A). Regarding gender, the association between RTI z-score and socioeconomic risk factors were higher among men for almost all socioeconomic indicators assessed. See the

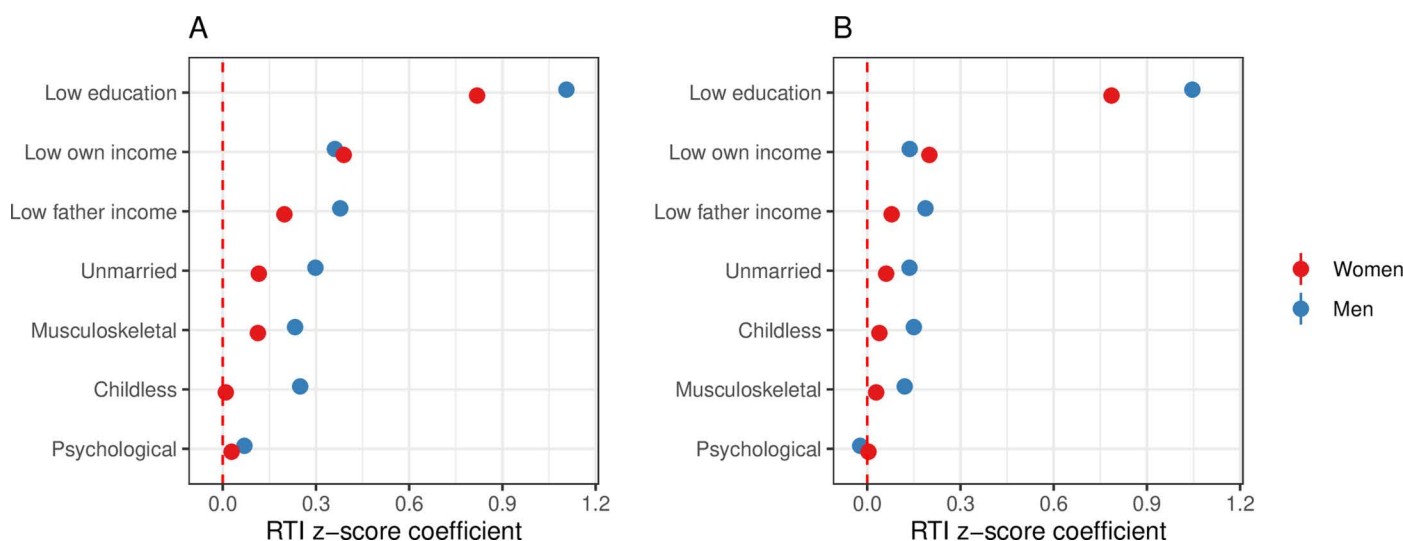

**Fig 2. Routine intensity risk factors by gender.** The figure displays regression coefficients from (A) bivariate and (B) multivariate regression models, with the RTI z-score as the outcome variable for all 45-year-olds between 2003 and 2018, separately by gender. The 95% confidence intervals are indicated by the error bars. Detailed regression tables are provided in the Supplementary Material (S3 Table).

Supplementary material for models using the Frey-Osbourne index (S5 Fig) and RTI 2019 z-score (S6 Fig).

## Explaining the increasing gender gap – attribution analysis

The gender differences in regression coefficients shown in Fig 2 could explain a gender gap in the RTI. However, to understand why this gap has increased over time, we also need to consider gender-specific changes in the prevalence of relevant risk factors. To explore this further, we conduct an attribution analysis to assess how the gender-specific development of each risk factor contributes to the overall change in the gap. This analysis was carried out in two steps. First, the gender-specific change in prevalence over the period 2003–2018 was calculated (Fig 3A). Second, to estimate the contribution, we multiplied these changes by the regression coefficients from the multivariate model (see Fig 2B) to estimate the relative impact of each risk factor to the evolving gender-specific risk of automation. These results are shown in Fig 3B.

The largest change in risk factor exposure was related to an increase in higher education (and correspondingly a decline in the share with low education; Fig 3A), especially among women (from 33 to 58 percent). There was also a substantial increase in the share of the sample that were unmarried, both among men (from 37 to 49 percent) and women (from 35 to 47 percent). Regarding the impact of the change in exposure on RTI z-score, i.e., when combining the strength of association (Fig 2B) with the change in exposure (Fig 3A), the strongest impact was found for education with an estimated impact on the RTI z-score of -0.117 (95% CI (-0.116)-(-0.118)) for men and -0.197 (95% CI (-0.195)-(-0.198)) for women.

## Distribution of structural risk across population groups

The estimated RTI z-scores from a multivariate model applied to the 2018 cohort of 45 years old individuals (birth cohort 1973) showed substantial variation across combinations of risk factors (Fig 4). While the predicted RTI z-scores were -0.75 [95% CI (-0.78)-(-0.72)] and -0.80 [95% CI (-0.83)-(-0.77)] among highly educated married men and women with children and

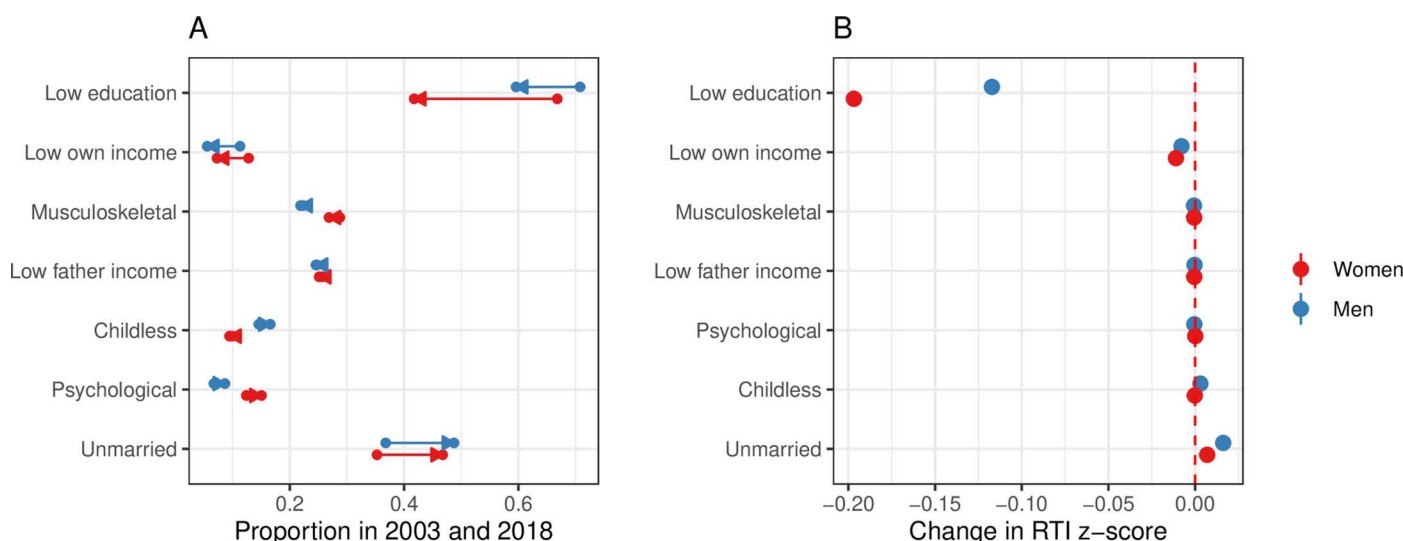

**Fig 3. Attribution analysis of routine intensity risk factors among 45-year-olds in 2003 and 2018, by gender.** (A) Share of 45-year-olds in 2003 and 2018 exposed to various risk factors. (B) Estimated impact of changes in exposure, calculated by multiplying the change in share (from Panel A) by the corresponding regression coefficient for each characteristic, based on a multivariate regression of the RTI z-score on all risk factors (see Fig 2B).

from high SES families, the corresponding RTI z-scores were 0.82 [95% CI 0.69–0.95] and 0.37 [95% CI 0.17–0.58] among less educated unmarried men and women without children from low SES families (see S4 Table and S5 Table for corresponding tables). To examine how this distribution has developed over time, we also estimated this model for the 2003 cohort (1958 birth cohort). The comparison across cohorts reveals that while the gender difference in automation risk declined among highly educated individuals, it has remained substantial among those of lower education. Hence, considering the increasing share of women taking higher education, it supports our finding that the increasing gender gap is due to a large shift in the share of women with university education (S7 Fig).

## Discussion

Our study has three main findings. First, the risk of automation was higher among men than women. This difference increased from 2003 to 2018. Second, the risk of automation was higher among individuals with low socioeconomic status and less social support in the form of a partner and children. Third, the main reason for the increasing gender gap was the growing gender differences in educational attainment. In summary, our study suggests that automation

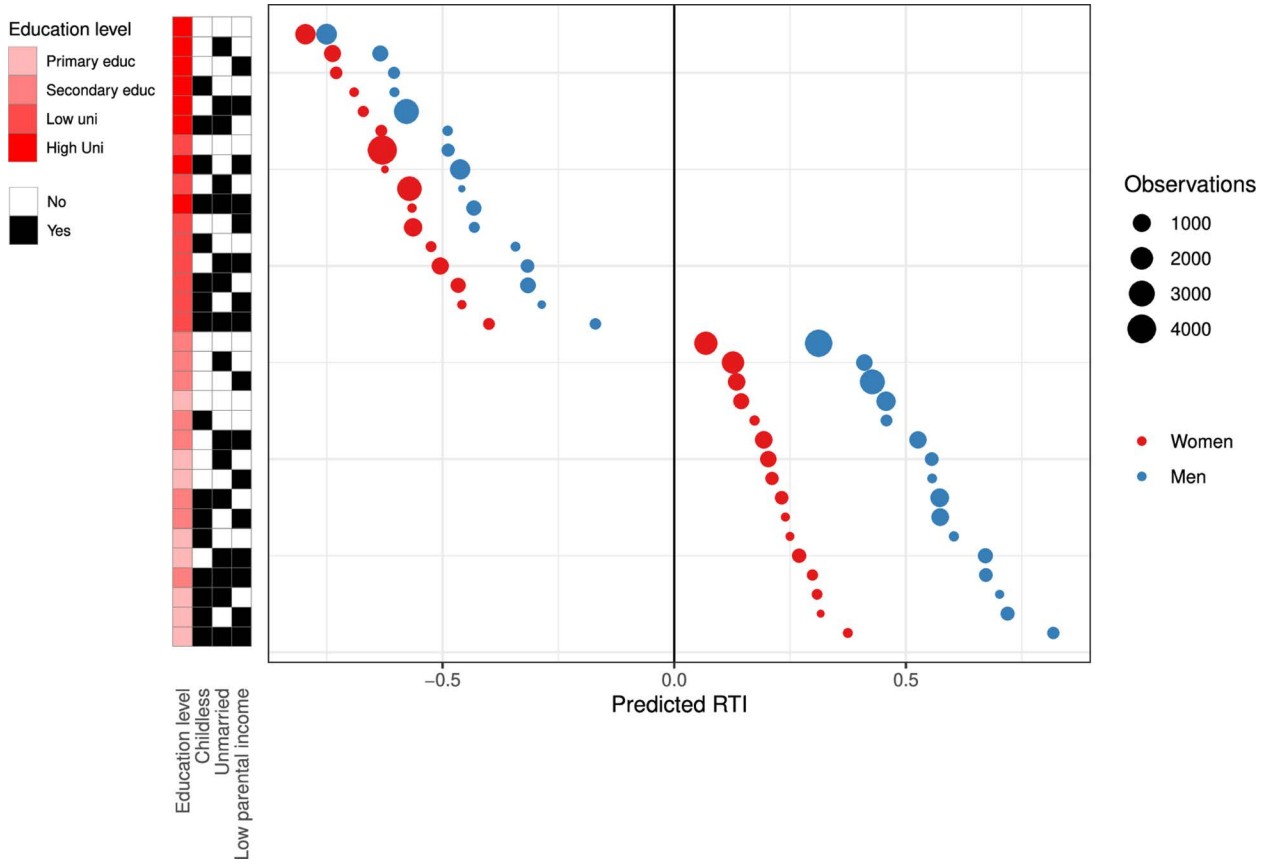

**Fig 4. Predicted RTI z-scores from multivariate regression by gender.** The figure presents predicted RTI z-scores based on gender-specific models estimated for the 1973 birth cohort (i.e., those aged 45 in 2018). The explanatory variables were indicators for educational level (Primary educ = primary education; Secondary educ = secondary education; Low uni = lower university (Bachelor's degree); High uni = higher university (Master's degree or higher)), indicators for childlessness (Childless), marital status (Married), and whether the father's average income rank between ages 40 and 50 was within the lowest quintile among men of the same age group (Low parental income). See Supplementary Material (S4 Table and S5 Table) for detailed tables and the average RTI z-scores for each group.

may exacerbate existing social inequalities and create a more unstable job market situation for men that may already struggle financially and socially. And, if disparities in economic and social opportunity are allowed to grow without mitigating policies, they have the potential to create social unrest and political polarization. At the same time, the nature of technological development changes rapidly, as evidenced by the introduction of large language models (LLMs) and multimodal models in recent years. These new developments may have further increased the scope of automation, potentially increasing the scope of routine tasks to also cover language-based tasks such as report writing and analysis.

Our study of Norway provides insights into automation risks within advanced economies, characterized by high educational attainment, strong norms of gender equality, and a well-developed technical infrastructure. A recent study covering 47 countries identified technology adoption and workforce skill levels as key factors driving cross-country variation in routine task intensity [32]. Therefore, developing countries, where larger shares of the population lack formal education, training, or work experience with information and communication technologies, were found to be more exposed to automation risks.

Gender disparities in automation risk have been examined across various regions, with mixed findings regarding which gender is more vulnerable [30,33–35]. We have found that the growing gender gap in the risk of automation can be attributed, in large part, to the increasing levels of education among women. It is interesting to note that this gap in education continues to widen, and Statistics Norway reported in 2019 that in many municipalities the share of highly educated women is more than twice that of men [36]. The gap found in this study likely reflects the significant rise in educational attainment among women in the Nordic countries over recent decades. However, the trend of increasing gender differences has, to the authors' knowledge, not yet been shown in population-wide data covering a long time period. As these developments are likely to continue, it is important for policy makers to understand their implications and carefully consider the best ways to address and mitigate the negative impacts.

Automation risk varies significantly across sectors, depending on the extent to which occupations consist of more routine tasks, or require interpersonal and emotional skills. The recent Future of Jobs Report by the World Economic Forum, finds that jobs within the care economy and technology sector are growing, while clerical secretarial workers are expected to see the largest declines [37]. In short, industries requiring a combination of emotional intelligence, complex problem-solving, and adaptability will remain challenging to automate in the foreseeable future. However, even within sectors assumed to grow over the coming decades, certain tasks (e.g., diagnostic imaging in healthcare or administrative work in education) are increasingly automated. In summary, the structure of the labor markets, educational trajectories chosen by different genders, cultural norms, technological readiness, industrial composition, will together determine how countries and genders, and other subgroups of the population, are differentially affected by technological developments over the coming decades.

## Strengths and weaknesses

The main strength of our study is that it covers the full population of employed 45-year-old individuals in Norway over a 15-year period across several important characteristics related to health, financial position, education, and family formation. Hence, in contrast to most other studies on this topic which uses survey data, our study does not suffer from selection bias. At the same time, our study has several weaknesses. First, the study has limited generalizability due to its focus on Norway, a country with unique characteristics related to gender roles, institutional environments, and industry structure. For example, Norway has an extensive welfare system that compensates 62.4 percent of the income in the case of job loss. Hence, the external validity of the results is limited, and likely most relevant for Northern European countries with similar

institutions and social norms. At the same time, similar gender-based occupational patterns are likely to exist in diverse geographic contexts. Consequently, the widening gender gap in automation risk and its association with social risk factors are likely present in countries also far from Norway. Second, the scope of the study is limited to what can be quantitatively measured in administrative registers and may not capture important experiential aspects related to technological development that could be measured with qualitative methods.

## Conclusion

The gender difference in the risk of automation has been increasing, with a particular high risk among lower educated men with fewer family ties. This potentially has significant and far-reaching negative impacts on individuals and society as it could exacerbate the concentration of economic, social and health exclusion in the coming decades. Given the increasing disparities identified in this study, policy makers should monitor the development in job market opportunities for various subgroups of the working-age population, and evaluate programs aimed at supporting individuals that are particularly negatively affected. For example, given the heightened risk faced by individuals in jobs that require less formal education, policymakers should consider implementing targeted upskilling programs that equip these workers—especially younger individuals—with skills that are in high demand and less likely to be at risk of automation. Last, provided the recent rapid developments in generative AI, future research should also explore how the risk distribution is evolving over time across age-groups, industries and social risk factors.

## Supporting information

**S1 File.  Constructing the routine intensity index (RTI).**
(DOCX)

**S1 Table.  Correlation matrix.**
(DOCX)

**S1 Fig.  The yearly average routine intensity index among 45-year-olds by gender, for different subgroups of the sample.** The figure shows the average RTI z-score, stratified by year for 45-year-olds (birth cohorts 1958–1973) by gender. The 95% confidence intervals are indicated by the error bars.
(TIF)

**S2 Table.  Gender-specific average RTI-z score among 45-year-olds, 2003–2018.**
(DOCX)

**S2 Fig.  Gender-specific yearly average Frey-Osborne index (FOI) among 45-year olds.** The figure shows the yearly average FOI, for 45-year-olds (birth cohorts 1958–1973) by gender. The 95% confidence intervals are indicated by the error bars.
(TIF)

**S3 Fig.  Gender-specific yearly average routine intensity among 45-year olds, RTI-2019 z-score.** The figure shows the yearly average RTI-2019 z-score, for 45-year-olds (birth cohorts 1958–1973) by gender. The 95% confidence intervals are indicated by the error bars.
(TIF)

**S4 Fig.  Gender-specific yearly average routine intensity among 45-year olds, raw RTI-2003 score.** The figure shows the yearly average RTI score, for 45-year-olds (birth cohorts 1958–1973) by gender. The 95% confidence intervals are indicated by the error bars.
(TIF)

**S5 Fig. Frey-Osborne index (FOI) risk factors by gender.** The figure displays regression coefficients from (A) bivariate and (B) multivariate regression models, with the FOI as the outcome variable for all 45-year-olds between 2003 and 2018, separately by gender. The 95% confidence intervals are indicated by the error bars.
(TIF)

**S6 Fig. Routine intensity risk factors by gender, RTI-2019 z-score.** The figure displays regression coefficients from (A) bivariate and (B) multivariate regression models, with the RTI-2019 z-score as the outcome variable for all 45-year-olds between 2003 and 2018, separately by gender. The 95% confidence intervals are indicated by the error bars.
(TIF)

**S3 Table. Results from bivariate and multivariate regression models.**
(DOCX)

**S7 Fig. Predicted RTI-score from a multivariate regression, for 2003 (birth cohort 1958).** The figure shows the predicted RTI-z-score from a model estimated on the 1958 birth cohort (45 years old in 2003) explaining RTI-z-score with four educational levels, a dummy for childless, a dummy for whether married and a dummy for whether father's income average income rank between ages 40 and 50 was within the lowest quintile of men of similar age.
(TIF)

**S4 Table. Predicted and average RTI z-scores for men (cf. Fig 4).**
(DOCX)

**S5 Table. Predicted and average RTI z-scores for women (cf. Fig 4).**
(DOCX)

## Author contributions

**Conceptualization:** Bjørn-Atle Reme, Ole Røgeberg, Bernt Bratsberg, Jonathan Wörn, Vegard Fykse Skirbekk.

**Data curation:** Bjørn-Atle Reme.

**Formal analysis:** Bjørn-Atle Reme.

**Methodology:** Bjørn-Atle Reme.

**Writing – original draft:** Bjørn-Atle Reme, Ole Røgeberg, Bernt Bratsberg, Vegard Fykse Skirbekk.

**Writing – review & editing:** Bjørn-Atle Reme, Ole Røgeberg, Bernt Bratsberg, Jonathan Wörn, Vegard Fykse Skirbekk.

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
