## [Decision Letter · Decision Letter 0]

17 May 2024

Dear Dr. Reme,

Thank you for submitting your manuscript to PLOS ONE. After careful consideration, we feel that it has merit but does not fully meet PLOS ONE’s publication criteria as it currently stands. Therefore, we invite you to submit a revised version of the manuscript that addresses the points raised during the review process.

We look forward to receiving your revised manuscript.

Kind regards,

Satabdi Mitra, M.D(Community Medicine )

Academic Editor

PLOS ONE

Journal Requirements:

"This work was financed by the Research Council of Norway through its Centres of Excellence funding scheme (project number 262700) and the project DIMJOB (project number 296297)."              

"We declare no competing interests."

We will update your Data Availability statement on your behalf to reflect the information

Reviewers' comments:

Reviewer's Responses to Questions

**Comments to the Author**

1. Is the manuscript technically sound, and do the data support the conclusions?

Reviewer #1: Yes

Reviewer #2: Yes

2. Has the statistical analysis been performed appropriately and rigorously?

Reviewer #1: Yes

Reviewer #2: Yes

3. Have the authors made all data underlying the findings in their manuscript fully available?

Reviewer #1: No

Reviewer #2: Yes

4. Is the manuscript presented in an intelligible fashion and written in standard English?

Reviewer #1: Yes

Reviewer #2: Yes

Reviewer #1: Comprehensive Data: The paper uses a rich dataset that includes multiple dimensions like education, income, employment, health, and demographics. This allows for a nuanced analysis.

Methodological Rigor: The paper employs both bivariate and multivariate regression models, allowing for a more robust understanding of the variables affecting the risk of automation.

Timely Topic: The paper addresses a very current and pressing issue—automation and its impact on the labour market, particularly in terms of gender disparities.

Attribution Analysis: The paper goes beyond merely identifying associations to actually estimating the relative importance of different risk factors over time.

Policy Relevance: The findings have significant policy implications, particularly for education and labour market policies.

Areas for Improvement:

Limited Generalizability: The study focuses solely on Norway, which might limit its applicability to other socio-economic contexts.

Data Sensitivity: The paper uses sensitive data that cannot be shared, which might limit the reproducibility of the study.

Complexity: The paper seems to be quite dense and might benefit from a simplified explanation of the key findings for a broader audience.

Lack of Qualitative Insights: While the paper is strong in its quantitative analysis, incorporating qualitative data could provide a more holistic view.

Future Projections: The paper could be strengthened by including future projections based on current trends, which would be valuable for policymakers.

Ethical Considerations: Given that the paper deals with sensitive data and has potential policy implications, a section discussing the ethical considerations would be beneficial.

Comparative Analysis: It might be useful to compare the situation in Norway with other countries to provide a more comprehensive view.

Reviewer #2: Strengths:

Comprehensive Data: The paper utilizes a robust dataset from Statistics Norway, which covers a wide range of demographic and socioeconomic indicators. This lends credibility to the findings and conclusions drawn from the analysis.

Relevance: The topic of automation and its impact on the labor market is timely and of significant importance, especially in the context of the ongoing technological revolution.

Detailed Analysis: The paper conducts a thorough analysis, including bivariate and multivariate regression models, to understand the relationship between automation risk and various socioeconomic indicators.

Clear Findings: The paper presents clear and well-structured findings, highlighting the increasing gender disparity in the risk of automation, especially among lower-educated men.

Policy Implications: The discussion section effectively ties the findings to potential policy implications, emphasizing the need for interventions to mitigate the negative impacts of automation.

Areas for Improvement:

Methodological Clarification: While the paper uses the RTI index and mentions the Frey-Osborne index, it would benefit from a more detailed explanation of why the RTI was chosen over other indices and the specific advantages it offers.

Comparative Analysis: It might be beneficial to compare the situation in Norway with other countries to provide a broader context and understand if the findings are unique to Norway or part of a global trend.

Potential Bias: The paper could address potential biases in the data, especially given that the data is from administrative registers. Are there any groups that might be underrepresented?

Future Implications: While the paper touches upon the potential negative impacts, a more in-depth exploration of the long-term implications of these findings would be valuable. For instance, what might be the societal consequences if these trends continue?

Recommendations: The conclusion section could benefit from specific policy recommendations based on the findings. For instance, what kind of educational or training programs might help mitigate the risks identified?

Recommendation:

Accept with Minor Changes.

The paper is well-researched, relevant, and provides valuable insights into the impact of automation on the labor market in Norway. However, addressing the areas of improvement mentioned above would strengthen the paper and make it more comprehensive.

**Do you want your identity to be public for this peer review?** For information about this choice, including consent withdrawal, please see our Privacy Policy

Reviewer #1: No

Reviewer #2: No

---

## [Decision Letter · Decision Letter 1]

25 Sep 2024

We look forward to receiving your revised manuscript.

Kind regards,

Satabdi Mitra, M.D(Community Medicine )

Academic Editor

PLOS ONE

Journal Requirements:

Reviewers' comments:

Reviewer's Responses to Questions

**Comments to the Author**

Reviewer #3: All comments have been addressed

Reviewer #4: (No Response)

2. Is the manuscript technically sound, and do the data support the conclusions?

Reviewer #3: Yes

Reviewer #4: Yes

3. Has the statistical analysis been performed appropriately and rigorously?

Reviewer #3: Yes

Reviewer #4: (No Response)

4. Have the authors made all data underlying the findings in their manuscript fully available?

Reviewer #3: Yes

Reviewer #4: Yes

5. Is the manuscript presented in an intelligible fashion and written in standard English?

Reviewer #3: Yes

Reviewer #4: Yes

Reviewer #3: 1. Introduction

- Clarification of research background: While the introduction already mentions the impact of technological changes on the labor market, further refinement of this background, particularly in the context of Norway, would be beneficial. For instance, specific Norwegian policies or trends in automation and globalization could be highlighted.

- Precision in stating research questions: In the introduction, it would be clearer to delineate the research questions, such as specifying which gender and socioeconomic disparities are to be investigated and how these disparities evolve over time.

2. Literature Review

- Exhaustiveness of literature citations: Although the paper references some relevant literature, incorporating more recent and authoritative sources to support the arguments, especially studies related to Norway or other European countries, would strengthen the review.

- Critical analysis of existing studies: When reviewing existing literature, a critical analysis of previous research should be conducted, identifying their limitations or research gaps, thereby clearly articulating the contribution of this study.

3. Methods

- Detailed description of data: When describing the data sources and analysis sample, provide a more detailed account of the data acquisition process, processing methods, and measures taken to ensure data quality.

- Explanation of analytical methods: For the RTI index and other statistical methods employed, offer more explanations and theoretical rationale to help readers better understand the applicability and advantages of these methods.

- Robustness checks: Although supplementary materials mention conducting robustness checks using different indicators, briefly mentioning these results in the main text would enhance the reliability of the conclusions.

4. Results

- Clarity in presentation of results: When presenting study results, use figures or tables to visually display the disparities between gender and socioeconomic status more intuitively.

5. Discussion and Conclusion

- Comprehensive discussion: In the discussion section, consider a more comprehensive examination of other potential explanatory factors, such as cultural and policy environments, to eliminate other potential confounding factors.

- Clarity of conclusions: The conclusion should explicitly state the main findings, contributions, and implications for policymakers and researchers.

- Future research directions: It would be beneficial to propose future research directions, such as further exploring differences across different age groups or industries, or studying the impact of technological changes on various social welfare indicators.

Reviewer #4: The manuscript titled “The Distribution of Technology-Induced Job Loss: Evidence from a Population-wide Study in Norway” explores a significant and timely topic—the impact of automation on job loss and socioeconomic disparities, with a particular focus on gender inequality. The study uses a large and comprehensive dataset from Norway, providing robust evidence to support its conclusions. Overall, the manuscript is well-written, but some revisions are required to improve clarity, expand the discussion, and address methodological details.

Strengths:

Using a comprehensive individual-level dataset allows for detailed analysis and lends credibility to the findings. The longitudinal nature of the data (2003-2018) enhances the robustness of the conclusions. The paper successfully highlights growing gender disparities in occupational routine intensity, particularly among individuals with low socioeconomic status, making a valuable contribution to labor economics and automation literature. The attribution analysis in explaining the increasing gender gap is a strong aspect of the paper, offering insight into the reasons behind these disparities.

Abstract Structure: The abstract should clearly outline the research objective, methodology, key findings, and implications. Consider adding a brief sentence at the end regarding policy implications or next steps to round it off.

1. Introduction: Ensure the introduction highlights the research gap more explicitly. This will better establish the context for the study and its relevance to the field. Furthermore, summarizing the hypothesis or key research questions can sharpen the focus.

Methodology Clarification:

1. Clearly specify the reasoning behind using the RTI index over other measures of automation risk like the Frey-Osborne Index, even though both are discussed. Include a brief comparison in the methodology section, not just in the results section, to improve clarity for readers.

2. Add more detail about how data access and linkage were done (e.g., anonymization procedures and encryption).

Data Presentation:

1. Ensure consistency in reporting the variables. The terms used for categories (e.g., education, income) should remain consistent throughout.

2. Consider providing more detailed tables or visual aids in the results section for critical variables to enhance clarity.

Discussion Section:

1. Address the limitations more extensively, particularly focusing on the external validity of the findings outside of Norway.

2. Expand on the implications of automation beyond just Norway to make the discussion more globally relevant. Including comparisons to other European countries or worldwide trends would enhance the broader significance of the findings.

Figures and Tables:

1. Ensure all figures and tables are fully labelled and include clear legends. Some figures (like those comparing gender over time) could benefit from a more detailed description in the figure caption to ensure they are standalone and clear to readers.

2. Ethical Considerations: Although the ethical aspects are addressed, adding a brief mention of any steps taken to minimize bias or stigmatization would further enhance transparency in research ethics.

Conclusions: Strengthen the conclusions by linking the findings more explicitly to potential policy responses or interventions. Mention any ongoing or future research that might address gaps left by this study. This would also help connect the study’s relevance to broader public and governmental policies.

Standardizing these sections will help align the paper more closely with scientific conventions and increase its clarity and impact.

**Do you want your identity to be public for this peer review?** For information about this choice, including consent withdrawal, please see our Privacy Policy

Reviewer #3: No

Reviewer #4: **Yes: ** RAPURU RUSHENDRAN

---

## [Author Response · Author response to Decision Letter 2]

20 Nov 2024

See separate attachment with response to reviewers letter.

---

## [Decision Letter · Decision Letter 2]

29 Dec 2024

Dear Dr. Reme,

We look forward to receiving your revised manuscript.

Kind regards,

Satabdi Mitra, M.D(Community Medicine )

Academic Editor

PLOS ONE

Journal Requirements:

Reviewers' comments:

Reviewer's Responses to Questions

**Comments to the Author**

Reviewer #5: (No Response)

Reviewer #6: (No Response)

2. Is the manuscript technically sound, and do the data support the conclusions?

Reviewer #5: Yes

Reviewer #6: Yes

3. Has the statistical analysis been performed appropriately and rigorously?

Reviewer #5: No

Reviewer #6: Yes

4. Have the authors made all data underlying the findings in their manuscript fully available?

Reviewer #5: Yes

Reviewer #6: Yes

5. Is the manuscript presented in an intelligible fashion and written in standard English?

Reviewer #5: No

Reviewer #6: Yes

Reviewer #5: Abstract line 1-28: The abstract seems unclear. Are you focusing on automation or gender differences? Please emphasize the most critical aspect of your manuscript.

Line 32: The phrase "increasing return to education" does not flow well. Consider breaking it into two sentences or simplifying it.

Line 33: What is OECD? Please provide the full form before introducing the abbreviation.

Line 33: By what year are 14% of jobs expected to be at risk?

Line 34: How will automation not cause changes in long-term unemployment? Please elaborate on this in the next sentence.

Line 40: What are the pre-existing inequalities? Consider adding examples such as gender disparities or differences between individuals with low and high education levels for clarity.

Line 41: Rephrase this sentence in a more scientifically acceptable manner.

Line 52: Does the negative impact affect low-skilled or high-skilled labor? Please clarify.

Line 82: Why did you choose individuals born between 1958 and 1973 and those who turned 45? Why were individuals under 45 not included?

Line 182: Include a statistical section in the manuscript with details on how the data was curated, organized, and analyzed. Specify the tools used, describe assumption testing conducted, and explain how the results can be interpreted from these tools (e.g., RTI and other risk tools). Mention the assumption testing conducted for regression analysis.

Line 269: The discussion section is too brief. Provide a more detailed description of the risk of automation, with references to automation risks in both developing and developed countries. Additionally, discuss which sectors are more or less likely to be affected.

Reviewer #6: Comments may be found in the attached document, but here are the same for convenience:

1. Please try and make the abstract structure. i.e., in sections Abstract, Materials and Methods, Results and Conclusion

If unstructured. Please quantitatively mention the “significant” findings of the result.

2. Regarding "low-education", Consider reframing

3. Please mention the sampling technique and then proceed to describe the details of the sample

4. Please clarify marital status a bit more clearly, or add the fallacy in the limitation: Separated couples, problem families may not always report to the register, and the register may not reflect the change in social contract

5.Regarding the final sentence in the Ethical considerations paragraph: Additionally, we

refrained from reporting on groups small enough to risk identifying individuals

“Small enough” is a vague term. Please mention what was the cutoff for considering a group “Small enough”

6. Large language models may be abbreviated as LLMs as the usage is more than once in the manuscript

7. Consider reframing sentences corresponding to 328 to 331.

**Do you want your identity to be public for this peer review?** For information about this choice, including consent withdrawal, please see our Privacy Policy

Reviewer #5: **Yes: ** Faizan Kashoo

Reviewer #6: **Yes: ** Vighnesh Devulapalli

---

## [Author Response · Author response to Decision Letter 3]

24 Jan 2025

Please see the attached response letter.

---

## [Editor Report · Decision Letter 3]

2 Mar 2025

The Distribution of Technology Induced Job Loss: Evidence from a Population-wide Study in Norway

PONE-D-23-25076R3

Dear Dr. Bjørn-Atle Reme,

We’re pleased to inform you that your manuscript has been judged scientifically suitable for publication and will be formally accepted for publication once it meets all outstanding technical requirements.

Kind regards,

Satabdi Mitra, M.D(Community Medicine )

Academic Editor

PLOS ONE
---

## [Editor Report · Acceptance letter]

PONE-D-23-25076R3

PLOS ONE

Dear Dr. Reme,

I'm pleased to inform you that your manuscript has been deemed suitable for publication in PLOS ONE. Congratulations! Your manuscript is now being handed over to our production team.

Kind regards,

on behalf of

Dr Satabdi Mitra

Academic Editor

PLOS ONE